# Chemical Identification of Specialized Metabolites from Sulla (*Hedysarum coronarium* L.) Collected in Southern Italy

**DOI:** 10.3390/molecules26154606

**Published:** 2021-07-29

**Authors:** Aldo Tava, Elisa Biazzi, Domenico Ronga, Mariella Mella, Filippo Doria, Trifone D’Addabbo, Vincenzo Candido, Pinarosa Avato

**Affiliations:** 1CREA Research Centre for Animal Production and Aquaculture, Viale Piacenza 29, 26900 Lodi, Italy; elisa.biazzi@crea.gov.it (E.B.); dronga@unisa.it (D.R.); 2Department of Pharmacy, University of Salerno, Via Giovanni Paolo II 132, 84084 Fisciano (SA), Italy; 3Department of Chemistry, University of Pavia, Viale Taramelli 12, 27100 Pavia, Italy; mariella.mella@unipv.it (M.M.); filippo.doria@unipv.it (F.D.); 4Institute for Sustainable Plant Protection, National Council of Research, 70125 Bari, Italy; trifone.daddabbo@ipsp.cnr.it; 5Department of European and Mediterranean Cultures, Environment and Cultural Heritage, University of Basilicata, Via Lanera 20, 75100 Matera, Italy; vincenzo.candido@unibas.it; 6Department of Pharmacy-Drug Sciences, University of Bari Aldo Moro, 70125 Bari, Italy; pinarosa.avato@uniba.it

**Keywords:** sulla, *Hedysarum coronarium* L., flavonoids, proanthocyanidins, saponins

## Abstract

Sulla (*Hedysarum coronarium* L.) is a biennal forage legume originated from the Mediterranean basin and used for animal feeding due to its high forage quality and palatability. Several species of *Hedysarum* have been considered for their nutritional, pharmaceutical, and biological properties, and different applications have been reported, both for human consumption and animal nutrition. Although a systematic investigation of the chemical constituents of *Hedysarum* spp. has been performed in order to provide chemotaxonomic evidences for the genus and to support the pharmacological application of several species within the genus, few data are available on the chemical constituents of *H. coronarium*, and only the content of condensed tannins and flavonoids in leaves has been previously reported. In the present paper, results from a detailed chemical analysis of the extracts from the leaves and flowers of *H. coronarium* grown wild in southern Italy are presented. Identification of the main specialized metabolites within the chemical classes of flavonoids, proanthocyanidins and saponins, is described, including considerations on their content in the two plant organs. Information acquired from this study expands the knowledge on *H. coronarium* as a source of valuable phytochemicals for different applications in human and animal health and nutrition.

## 1. Introduction

The genus *Hedysarum* L. is the largest genus of the tribe Hedysareae (Papilionoideae, Fabaceae) and consists of about 160 species, widely distributed in the temperate northern hemisphere. Several species from this genus are considered for their nutritional, pharmaceutical, and biological properties, and different applications have been reported, both for human consumption and animal nutrition. In China, several species belonging to this genus have a long history of use as restorative foods and in traditional Chinese medicine (TCM); the dried roots of *H. polybotrys* Hand.-Mazz. (*radix Hedysari*) and *H. multijugum* Maxim are employed for the treatment of various diseases, such as the treatment of palpitation and chronic nephritis or to improve the immune system [1,2,3]. Moreover, hypocholesterolemic and laxative activity have been reported for the flowers [4]. To the best of our knowledge, specific studies on the medicinal properties of *H. coronarium* are limited, though extracts from this species with a high content of phenolics have been proposed as suitable material to develop dermatological and cosmeceutical products [5].

In the Mediterranean region, several native *Hedysarum* spp. are present; among them, sulla, *H. coronarium* L., is a cultivated biennal legume used for animal feeding, either for its high forage quality and palatability [6,7] and beneficial effects on animals due to the presence of several metabolites, including proanthocyanidins, or condensed tannins, whose overall content may be influenced by the employed agronomic practices [8,9,10]. The potential benefits attributed to sulla condensed tannins include their ability to bind with dietary proteins and reduce the rumen proteolysis, which can improve protein utilization in ruminant animals. Condensed tannins have also been reported to have anthelmintic effects on gastrointestinal nematodes and enhanced host immune response, with positive consequences for animal performance [11,12,13,14,15].

A systematic investigation of the chemical constituents of *Hedysarum* spp. has been performed in order to provide chemotaxonomic evidence for the genus [16] and to support the pharmacological application of several species within the genus [3,5,17]. The phytochemical study led to the identification of several specialized metabolites such as flavonoids (flavones, flavanones, flavanols, isoflavones, pterocarpanes, and chalcones) [16,18,19,20,21,22,23], xanthones [3], tannins [21], coumestans [16,24], benzofurans [16,20,25], and oleanane saponins [1,19].

To the best of our knowledge, only one paper has been published on the chemical composition of *H. coronarium* aerial parts [21]. The plants of sulla analyzed in this work were collected in New Zealand, and the composition of condensed tannins and some phenolic constituents from the plant leaves harvested in different seasons were described.

In the present paper, results from a detailed chemical analysis of the extracts from the leaves and flowers of *H. coronarium* grown wild in southern Italy are reported. Identification of the main specialized metabolites within the chemical classes of flavonoids, proanthocyanidins and saponins, is described, including considerations on their content in the two plant organs. Information acquired from this study expands the knowledge on *H. coronarium* as a source of valuable phytochemicals for different applications in human and animal health and nutrition.

## 2. Results and Discussion

### 2.1. Identification and Quantitation of Flavonoids

Phenolics occurring in the extract from the flowers and leaves of *H. coronarium* were separated by HPLC/DAD method (see Section 3), and UV spectra were recorded by a photodiode array detector. All the components of each extract were also analyzed by LC/MS, and mass spectra in positive and negative ion mode were collected. The HPLC chromatograms of flowers and leaf extracts are shown in Figure 1. Identification of compounds was performed considering their retention times; UV absorptions and mass spectral data were also compared with those of available standard compounds, as well as with previously identified constituents of *Hedysarum* spp. And other data reported in the literature [20,21,22,26,27,28].

UV data (Table 1) indicated the presence of two distinct groups of metabolites: one group was characterized by two maxima of absorption in the range of 240–280 nm (Band II, A-ring benzoyl system) and 330–380 nm (Band I, B-ring cynnamoyl system), see Figure 2, with weak absorptions (*sh)* around 299 nm, consistent with the structure of flavonols or flavones; the other sub-type typically showed a strong absorption of Band II (245–270 nm) and only a low intensity shoulder of Band I, thus supporting the presence of isoflavone components [29]. In particular, small shifts in their λ_max_ allowed us to distinguish flavonol derivatives of kaempferol (264, 299*sh*, 330*sh*, 364 nm), quercetin (256, 269*sh*, 300*sh*, 370 nm), isorhamnetin (255, 266*sh*, 298*sh*, 355 nm), and myricetin (255, 270*sh*, 299*sh*, 375 nm). Moreover, UV data also suggested the presence of isoflavones derivatives of formononetin (250, 303*sh* nm) and afrormosin (260, 325*sh* nm). Full UV data for each component in the two extracts (flowers and leaves) of *H. coronarium* are reported in Table 1. The above considerations were further confirmed by the observed mass fragmentation pattern (Table 1); that is, components of the extracts were characterized by [Aglycone + H]^+^/[Aglycone − H]^−^ ions at *m*/*z* 287/285, 303/301, 317/315, 319/317, 269, and 299, corresponding to kaempferol, quercetin, isorhamnetin, myricetin, formononetin, and afrormosin, respectively. In addition, diagnostic fragments deriving from the loss of sugars (−162 Da, loss of glucose or galactose; −146 Da, loss of rhamnose) from the pseudomolecular ions indicated that the identified compounds were all *O*-glycosides, present as mono-, di-, and tri-glycosides, with rutinose (rhamnosyl-(1→6)-glucose, −308 Da) as the common disaccharide moiety. Finally, mass spectra also allowed the identification of some acylated glycosyl flavonoids; in particular, the loss of 42 Da and the loss of 44 and 86 Da from the pseudomolecular ions in the negative ion mode were indicative of the presence of an acetyl and a malonyl group in the molecule, respectively.

The most abundantly identified flavonoid glycosides and their content in the two different plant organs are reported in Table 1, and their chemical structures are reported in Figure 2. The major constituents of both flower and leaf extracts were glycosyl derivatives of the flavonols kaempferol and quercetin and the isoflavone formononetin. Minor amounts of myricetin, isorhamnetin, and afrormosin glycosides were also identified in both extracts. The ramified trisaccharide chains in compounds **2**, **3**, and **8** were identified based on fragmentation pattern that, with a serial loss from the corresponding molecular ion [M + H]^+^ of two rhamnose units, formed the intermediate ion that in turn, with a further loss of an hexose unit (glucoside), gave the corresponding aglycone ion. This indicated that the hexose was attached to the aglycone moiety and the two rhamnose units were attached to the hexose rather than the aglycone. These compounds were previously detected in *H. coronarium* extracts [21] and also characterized in *H. carnosum* [22].

A series of 3-*O*-rutinoside derivatives were characterized, including myricetin (**1**), two quercetin (**4** and **6**), two kaempferol (**9** and **13**), and two isorhamnetin (**14** and **15**) derivatives. The 7-*O*-rutinoside of 5,7-dihydroxy-3′,4′,5′-trimethoxyflavone (**16**) was also characterized. Rutin (**6**) and nicotiflorin (**9**) were unambiguously identified based on retention time and MS spectral data of authentic reference standards. Flavonol 3-*O*-rutinosides have been previously reported as constituents of the phenolic fraction of *Hedysarum* spp. [21,22].

Flavonol monoglycosides were also detected in very low amounts, including tricin-7-*O*-galactoside (**7**), kaempferol-3-*O*-galactoside (**10**), and the malonyl and the acetyl derivatives of quercetin-3-*O*-glucoside (**11** and **12**, respectively). Among isoflavones, formononetin-7-*O*-glucoside (**20**), afrormosin-7-*O*-glucoside (**21**), and their malonyl derivatives (**22** and **23**) were also identified in both the plant organs. The finding of these isoflavones is in agreement with previous reports from several *Hedysarum* spp., including *H. coronarium* [16,21,22,30].

Flavonoids were quantitated by an external standard method using rutin and nicotiflorin for quercetin and kaempferol glycosides, respectively, while ononin was used to quantitate isoflavones. Results are reported in Table 1. As shown, flowers contain a higher amount of phenolics, quoted as 6.26 ± 0.75 mg/g DM, compared to leaves, in which they are quoted as 2.72 ± 0.38 mg/g DM. Based on the presence of a higher amount of rutin (**6**) in both plant organs (1.15 ± 0.23 mg/g and 0.71 ± 0.07 mg/g DM in flowers and leaves, respectively), flowers are characterized by the presence of a higher amount of the 3-*O*-(2”,6”-di-*O*-rhamnosyl)glucoside flavonol derivatives **2** and **3** (1.60 ± 0.38 mg/g DM and 1.15 ± 0.12 mg/g DM, respectively), while leaves are instead characterized by a higher amount of kaempferol-3-*O*-rutinoside (**9**) (0.25 ± 0.04 mg/g DM) and isorhamnetin-3-*O*-rutinoside (**14**) (0.38 ± 0.06 mg/g DM). The isoflavone formononetin-7-*O*-glucoside (**20**) and its 6”-*O*-malonyl derivatives (**22**) were also found in a relatively higher amount in both plant organs (see Figure 1 and Table 1). The branched trisaccharide chain, present in compounds **2**, **3**, and **8**, seems to be characteristic and typical of glycosilated flavonols from flowers, while the two quercetin derivatives, quercetin-3-*O*-malonyl glucoside (**11**) and quercetin-3-*O*-acetyl glucoside (**12**), were only detected in the leaf extract.

### 2.2. Identification and Quantitation of Saponins

The saponin fractions from the leaves and flowers of sulla were evaluated by HPLC/DAD and LC/MS method (see Section 3), and UV and mass spectra in negative ion mode were collected. Saponins were identified based on their retention times, and mass spectral data were compared with those of previously purified and identified standard compounds, as well as with other data from the literature [31,32,33,34]. Identification of DDMP (2,3-dihydro-2,5-dihydroxy-6-methyl-4H-pyran-4-one) derivatives of saponins was also achieved based on their UV spectra, with a characteristic strong absorption band at λ_max_ 270 nm [31]. Figure 2 shows the chemical structures of soyasaponin I and its DDMP derivative as the most abundant saponin detected in the sulla extracts.

To confirm its presence in the two extracts, soyasaponin I was purified by chromatographic method (see Section 3) and analyzed by NMR experiments. By comparison of the obtained data with those of previously purified and identified soyasaponin I and data from literature [1,19,35], all the NMR signals were well attributed (data not shown), confirming the structure of this compound.

All saponins were quantitated by an external standard method using a previously purified and identified soyasaponin I [35], and results are reported in Table 2. Flowers contained a higher amount of saponins (6.28 ± 0.30 mg/g DM compared to 2.88 ± 0.23 mg/g DM in the leaves extract). Soyasaponin I (**24**) and its 22-*O*-DDMP derivative (**28**) were the most abundant saponins detected in both plant organs, quoted as 3.79 ± 0.25 and 1.74 ± 0.30 mg/g DM, respectively, in flowers and 1.63 ± 0.17 and 1.02 ± 0.06 mg/g DM, respectively, in leaves. Other saponins, such as soyasaponin **25** and its 22-*O*-DDMP derivative (**29**) and two saponins of soyasapogenol E (compounds **26** and **27**) were detected in lower amounts (Table 2).

Soyasaponins are specialized metabolites present in several plant species, including some belonging to the Leguminosae family. In the genus *Hedysarum*, these compounds have not been particularly studied and their presence have been reported only in the roots of *H. multijugum* and *H. polybotris* [3], as the dried roots of this last plant species are largely used in TCM. To the best of our knowledge, this is the first report on the presence of these saponins in *H. coronarium*.

### 2.3. Identification and Quantitation of Proanthocyanidins

The amount of proanthocyanidins in both the examined plant organs of *H. coronarium* were quantitated by the butanol/HCl method using a previously purified tannin fraction from sulla as the reference compounds (see Section 3). Flowers contained higher amount of proanthocyanidins, quoted as 152.2 ± 2.2 mg/g DM, compared to leaves in which they account for 138.8 ± 1.5 mg/g DM. These results agree with those from the literature, which reported a variability in proanthocyanidin concentration in *H. coronarium* aerial parts related to the plant organs, with flowers showing the highest concentration [21,36,37].

The proanthocyanidins from *H. coronarium* leaves were extracted and purified according to a well-established procedure [21,38,39,40]. Condensed tannins were extracted from the plant material with acetone/water mixture and then purified and fractionated using chromatographic methods. By Sephadex LH 20 column, the lower molecular weight phenolics were firstly eluted by aqueous methanol (1:1), and then the condensed tannins were obtained by elution with aqueous acetone (3:7). Finally, from the purified total tannin extract, two fractions were then separated by means of RP-18 column chromatography (see Section 3). Investigation of the chemical structure of these polymeric fractions was performed by their acid-catalyzed depolymerization in the presence of benzyl mercaptan [21,38,39,40,41] and by NMR methods [40,42,43,44]. Thiolytic degradation of proanthocyanidins provides good yields of cleavage products [41], with low levels of degradation and epimerization [38]. In thiolysis reactions, all the extender units of flavan-3-ol polymers react with benzyl mercaptan to form the corresponding benzyl thioether. Only the terminal units are released as the free flavan-3-ol.

The thiolysis reaction products of our samples were analyzed by HPLC/DAD to determine the nature of each individual unit in the polymer, the procyanidin (PC) to prodelphinidin (PD) ratio, and the mean degree of polymerization (mDP). The mean composition of the terminal units has been determined from the ratio of the released monomers, the mean composition of the extender units in the polymer chain has been determined from the ratio of benzyl thioether adducts, and the mDP from the ratio of monomers to extender units [21,38,39,40,41]. Results of thiolytic degradation on purified total condensed tannins and the two fractions from the RP18 column are summarized in Table 3. The thiolysis products indicated a consistent mean polymer chain length for all the proanthocyanidin fractions, with a range from 15 to 25 mDP. In all fractions, catechin (C) was the dominant terminal unit (51–76%), followed by gallocatechin (GC) (19–30%), while epigallocatechin (EGC) and GC were the dominant extender units, detected as 47–57% and 32–38% of the total extenders, respectively. The predominance of GC and EGC is a characteristic of prodelphinidin predominant condensed tannins; the prodelphinidin to procyanidin overall ratio remains fairly constant, from 85:15 to 90:10 (Table 3). The predominance of EGC among the extender units indicated that the 2,3-*cis* stereochemistry was predominant in all the proanthocyanidin fractions. The predominance of C and GC among the terminal units indicated the 2,3-*trans* stereochemistry in all the tannin fractions. Thus, sulla tannins were found to be predominantly of the prodelphinidin type, dominated by *cis* extender units and *trans* terminal units. These data are in agreement with those previously reported for the tannins of *H. coronarium* [21].

^13^C-NMR studies were also performed in order to obtain more information on the chemical structure of sulla polymeric proanthocyanidins [43]. The ^13^C-NMR spectrum of total tannins is shown in Figure 3. Several large signals were observed, indicating polymeric compounds; nevertheless, in spite of the broad nature of the peaks, the resolution was sufficient to allow for some structural interpretation. Based on literature data [40,42,43,44], all signals were attributed as described in Figure 3. The ^13^C-NMR spectrum shows distinct signals at 146 ppm (C3′ and C5′), 135 ppm (C4′), and 108 ppm (C2′ and C6′), which are assignable to the carbons of the catechol B-ring of prodelphinidin units (gallocatechin/epigallocatechin). The corresponding procyanidin carbon resonances are registered at 143 ppm (C3′ and C4′), 120 ppm (C6′), and 116 ppm (C2′ and C5′). The region from 70 and 90 ppm is diagnostic for the 2,3-stereochemistry of the C ring. While the signal for the extender C3 of both *cis* and *trans* isomers occurs at 73 ppm, the corresponding C2 signal for the *cis* and the *trans* forms are well resolved; the *cis*-C2 signal occurs at 77 ppm and the *trans*-C2 signal occurs at 84 ppm. The relatively large signal at 77 ppm as compared to that at 84 ppm provides a clear indication of the preponderance of the *cis* stereochemistry among the extender units. The carbon chemical shift of C3 of the extender units occurs at 73 ppm, while the same carbon of the flavanol terminal unit occurs at 68 ppm. From the intensity of these signals, the mDP could be estimated to be about 18–20 DP. These results are in agreement with those obtained from the thiolytic method.

## 3. Materials and Methods

### 3.1. Plant Material

Wild plants of *H. coronarium* L. were collected from Irsina, Matera, Italy (40°47′15″ N, 16°12′57″ E, 290 a.s.m.) at the beginning of May when plants were at the flowering stage. The plant samples were deep-frozen and immediately brought to the laboratory. Leaves and flowers were separated, freeze dried, finely powdered, defatted with chloroform, and then used for the subsequent extractions.

### 3.2. Extraction and Purification of Flavonoids

The defatted plant material (100 mg) was extracted with 5 mL of 80% MeOH under stirring overnight. After centrifugation at 3000× *g*, the supernatant was separated, and the residue re-extracted with 5 mL of 80% MeOH. The combined extracts were diluted to 30% MeOH and applied onto a RP18 cartridge (400 mg, Merck, Milano, Italy) preconditioned with 30% MeOH. The cartridge was washed, first with 30% MeOH (5 mL) to remove sugars and then with 80% MeOH (5 mL) to elute flavonoids. Both the extracts obtained from leaves and flowers were evaporated to dryness under reduced pressure at 40 °C, re-dissolved in 1 mL of 80% MeOH, and then used for the chemical analyses. Three independent purified extracts were prepared from each plant sample (leaves and flowers) and separately analyzed by HPLC/DAD analysis and LC-ESI-MS.

### 3.3. Extraction and Purification of Saponins

Samples of the defatted plant material (100 mg) from leaves and flowers were separately extracted with 5 mL of 30% MeOH under stirring overnight. After centrifugation at 3000× *g*, the supernatant was separated, and the residue extracted again with 5 mL of 30% MeOH. The combined extracts were applied onto a RP18 cartridge (400 mg, Merck, Milano, Italy) preconditioned with 30% MeOH. The cartridge was washed, first with 50% MeOH (5 mL) to remove sugars and some phenolics and then eluted with 90% MeOH (5 mL) to obtain saponins. Each extract was evaporated to dryness under reduced pressure at 40 °C, re-dissolved in 1 mL of 90% MeOH, and then used for the following analyses. Three independent extractions and purification steps were performed on each sample and separately used for HPLC/DAD and LC-ESI-MS analyses.

Soyasaponin I (**24**) was also purified from the crude saponin extract from flowers by open column chromatography, according to a standard procedure [35]. A sample of defatted flowers (1 g) was extracted with 50 mL of 30% MeOH under stirring overnight. After centrifugation at 3000× *g*, the supernatant was separated and treated with few drops of diluted hydrochloric acid overnight at room temperature to completely hydrolyse the DDMP derivatives. The solution was loaded onto a C18 column (Lichroprep RP-18, 50 × 20 mm, 40–63 μm, Merck, Milano, Italy), equilibrated with 30% MeOH and washed with the same solution to completely remove the acid. Elution of saponins was performed with increasing MeOH concentration, and fractions were checked by silica gel TLC plates, developed with ethyl acetate/acetic acid/water (7:2:2). Spots were visualized by spraying the TLC plates with Liebermann–Burchard reagent (MeOH/acetic anhydride/sulphuric acid, 10:1:1 *v*/*v*). Pure soyasaponin I (**24**) (7.2 mg) was obtained with MeOH of about 65% concentration.

### 3.4. LC-ESI-MS Analysis and Identification of Flavonoids and Saponins

Identification of flavonoids and saponins in the leaves and flowers extracts of sulla was based on their retention times and mass spectral data (positive and negative mode) with those of available standard compounds, as well as with previously identified constituents of *Hedysarum* spp. [19,20,21,22,26] and other data reported in the literature [28,31,32,33,34,45]. For identification of flavonoids and DDMP derivatives of saponins, UV spectra acquired by HPLC/DAD were also used.

For ESI/MS-MS, a Jasco UPLC system equipped with a binary pump system and photo diode array detector (Jasco Chemstation ChromNAV, Jasco, Mary’s Court, Easton, MD, USA), and coupled to a Thermo LTQ (linear ion trap mass detector) with an electrospray ionization (ESI) source was used. All data were acquired and processed using Thermo Xcalibur Qual Browser software. Chromatographic runs were carried out with an Acquity UPLC BEH C18 column (50 mm × 2.1 mm i.d., 1.7 µm particles, 13 nm pore size, Phenomenex, Torrence, CA, USA). Before injection, all solutions were filtered by syringe filter with a nylon membrane (0.2 μm, Nalgene, Milano, Italy). Five microliters of methanolic solutions (1 mg/mL) of all samples were injected. The mobile phase for all the UPLC analyses consists in the following solvents: A, CH_3_CN/0.1% HCOOH and B, H_2_O/0.1% HCOOH. Elution was performed with a flow rate of 0.3 mL/min. The eluates were spectrophotometrically checked at different wavelengths (from 210 to 390 nm).

Flavonoids were eluted with the following gradient method: 5% A (5 min isocratic condition) to 25% A in 30 min, to 70% A in 40 min, and then to 100% A in 50 min. Mass-spectrometry data were acquired in a positive and negative ion mode. For saponins, the following gradient method was used: 20% A (5 min isocratic condition) to 30% A in 30 min, to 80% of A in 20 min, and then to 100% of A in 30 min. For MS detection, negative ESI was used as the ionization mode.

All the analyses were carried out using an ESI ion source with the following settings: capillary voltage, 3100 V; Sheath gas (He), Aux gas (He), and Sweep gas (He) heated at 275 °C and introduced with a source heater temperature of 80 °C. Full scan spectra were acquired over the range of 100–2000 *m*/*z*. Automated MS/MS was performed by isolating the base peaks (molecular ions) using an isolation width of 2.0 *m*/*z*, normalized collision energy of 25 V, threshold set at 500, and ion charge control on, with max acquire time set at 300 ms.

### 3.5. HPLC/DAD Analysis and Quantitation of Flavonoids and Saponins

All extracts were analyzed by HPLC using a Perkin–Elmer chromatograph equipped with an LC 250 binary pump and a DAD 235 detector (Perkin Elmer, Milano, Italy). Separation was performed on a 250 mm × 4.6 mm i.d., 5 μm, Discovery HS-C18 column (Supelco, Milano, Italy) with a mobile phase, consisting of solvent A, CH_3_CN/0.05% CF_3_COOH, and solvent B, H_2_O/1% MeOH/0.05% CF_3_COOH. Before injection, all solutions were filtered by syringe filter with a nylon membrane (0.2 μm, Nalgene, Milano, Italy). Twenty microliters of methanolic solutions (1 mg/mL) of all samples were injected, and compounds were eluted at 1.0 mL/min.

For flavonoids, chromatographic runs were carried out under gradient elution from 10% (15 min isocratic condition) to 30% of A in 30 min and then to 80% of A in 10 min. Detection was by UV monitoring at 260 nm. For saponins, chromatographic runs were carried out under gradient elution from 20% (5 min isocratic condition) to 30% of A in 30 min and then to 80% of A in 20 min. Detection was by UV monitoring at 215 nm.

Individual components were quantitated against reference standards; concentration of kaempferol and quercetin glycosides were calculated using the standard curve of nicotiflorin and rutin (Sigma–Aldrich, Milano, Italy), respectively, while the concentration of isoflavones were calculated using the standard curve of ononin (Sigma–Aldrich, Milano, Italy), respectively. Concentrations of saponins were calculated using the standard curve of a previously purified and identified soyasaponin I [35].

The external standard method was employed for quantitation. A seven-level calibration curve (0.0012–0.4 mg/mL for flavonoids and 0.225–1.5 mg/mL for soyasaponin I) was created for all standards. The correlation coefficient (r^2^) of the standard curve in the linear plot was: r^2^ = 0.998 (y = 0.000035x − 7.9) for nicotiflorin, r^2^ = 0.999 (y = 0.000033x − 2.7) for rutin, r^2^ = 0.999 (y = 0.000016x − 0.038) for ononin, and r^2^ = 0.995 (y = 0.00046x − 16.3) for soyasaponin I, indicating a good linearity between peak areas and concentration within the tested concentration range. The precision of the adopted HPLC method was determined by calculation of the intra-day % RSDs of retention times (0.22% to 1.51% for nicotiflorin, 0.85% to 1.21% for rutin, 0.15% to 1.25% for ononin, and 0.36% to 1.84% for soyasaponin I) and peak areas (1.58% to 2.84% for nicotiflorin, 1.12% to 3.36% for rutin, 1.57% to 2.95% for ononin, and for 2.31% to 4.25% soyasaponin I). Calculated % RSDs for inter-day retention times were: 0.56% for nicotiflorin, 0.64% for rutin, 0.37% for ononin, and 0.97% for soyasaponin I, while for peak areas were 1.35% for nicotiflorin, 3.95% for rutin, 2.21% for ononin, and for 4.98% soyasaponin I. The limit of detection (LoD) was 0.07 μg/mL for nicotiflorin, 0.15 μg/mL for rutin, 0.08 μg/mL for ononin, and 0.28 μg/mL for soyasaponin I. The limit of quantification (LoQ) was 0.12 μg/mL for nicotiflorin, 0.18 μg/mL for rutin, 0.15 μg/mL for ononin, and 0.36 μg/mL for soyasaponin I. All HPLC analyses for identification and quantitation of compounds were run in triplicate.

### 3.6. Quantitation of Proanthocyanidins by the Butanol/HCl Assay

Proanthocyanidin (condensed tannins) content was assessed by the butanol/HCl method [46]. Leaf samples (50 mg) were extracted with 3 mL of acetone/water (7:3 *v*/*v*) containing 1 g/L ascorbic acid, under stirring overnight at room temperature. The samples were then centrifuged at 2500× *g*, and the supernatant were used for tannin determination. A sample of 0.5 mL of plant extract was added to 3 mL of butanol/HCl (95:5 *v*/*v*), followed by 0.1 mL of a 50 g/L solution of NH_4_Fe(SO_4_)_2_· 12H_2_O in HCl 2M. The samples were incubated at 95 °C for 40 min to develop a red coloration and then evaluated at 550 nm by Lambda 365 UV/Vis spectrophotometer (Perkin Elmer, Milano, Italy). All samples were extracted in triplicate and results expressed in mg/g dry matter as the mean of three independent analyses ± standard deviation. A purified condensed tannin fraction from sulla was used as the reference compound for the quantitative evaluation.

### 3.7. Extraction and Purification of Proanthocyanidins

Dried sulla leaves (250 g) were extracted with 750 mL of acetone/water (7:3 *v*/*v*) under stirring for 30 min. The extract was filtered using a paper filter and the plant material extracted again in the same conditions. The combined extracts were then concentrated in vacuo at 40 °C to remove acetone, and the obtained aqueous solution was defatted with CH_2_Cl_2_ (3 × 200 mL). From the aqueous layer, the residual CH_2_Cl_2_ was eliminated under vacuum, and then the solution was freeze dried to yield 19.7 g of brown solid of crude proanthocyanidins.

Five grams of the condensed tannin extract was dissolved in 50 mL of MeOH/H_2_O (1:1), filtered under vacuum using a paper filter, and applied onto a 25 × 3.5 cm Sephadex LH20 (100 μm, Pharmacia, Milano, Italy) column equilibrated with MeOH/H_2_O (1:1). A first elution with MeOH/H_2_O (1:1, 1 L) allowed the removal of sugars, lower molecular weight phenolics, and other contaminants, while acetone/H_2_O (7:3, 2.5 L) allowed the elution of tannins. Acetone was removed under vacuum, and the aqueous fraction was freeze dried to obtain 3.2 g of crude tannins as a brownish powder. An amount of 2.5 g of crude tannins were dissolved in water and applied onto RP-18 open column chromatography (Lichroprep RP-18, 50 × 20 mm, 40–63 μm, Merck, Milano, Italy). Elution was performed with 100 mL aliquot parts of acetone/water solutions from 10 to 50% acetone. All eluted fractions were analyzed by HPLC/DAD monitoring at 280 nm. The fractions with the same chromatographic profile were pooled together, concentrated under vacuum, and freeze dried. Two fractions were obtained: fraction 1 (1.1 g) and fraction 2 (0.7 g).

### 3.8. Thiolysis of Proanthocyanidins

Thiolysis was carried out on freeze dried samples according to [40]. An amount of 100 μL of proanthocyanidin solution (4 mg/mL in MeOH) was placed in a corked vial and combined with 100 μL of 3.3% HCl in MeOH and 200 μL of 5% benzylmercaptan in MeOH. The solution was heated to 40 °C for 30 min in a heating block and then cooled to room temperature. An amount of 200 μL of dihydroquercetin (0.05 mg/mL in MeOH) was added as the internal standard, and samples were immediately analyzed by HPLC.

A Perkin–Elmer chromatograph equipped with an LC 250 binary pump and a DAD 235 detector was used. Separation was performed on a 250 mm × 4.6 mm i.d., 5 μm, Discovery HS-C18 column (Supelco, Milano, Italy) with a mobile phase, consisting of solvent A, CH_3_CN/0.05% CF_3_COOH, and solvent B, H_2_O/1% MeOH/0.05% CF_3_COOH. Chromatographic runs were carried out under gradient elution from 10% (10 min isocratic condition) to 30% of A in 20 min, to 60% of A in 30 min and then to 90% of A in 15 min. Detection was by UV monitoring at 280 nm. Concentration of terminal flavan-3-ol units and extender flavan-3-ol thiol adducts were estimated relative to dihydroquercetin as the internal standard, calculating the response factors for catechin/epicatechin, gallocatechin/epigallocatechin, and their corresponding benzothioethers at 280 nm, in accordance with [38]. A linear response of flavan-3-ols (conc/int std conc) vs. peak area (area/int std area) was obtained between 0.06 and 1.3 mg/mL, and the HPLC condition described above allowed a practical measurable sensitivity of 1.2 ng/μL flavan-3-ol per injection.

### 3.9. NMR Analyses

^1^H- and ^13^C-NMR were measured on a Bruker AV-300 spectrometer at the operating frequencies of 300.13 and 75.13 MHz, respectively (Bruker, Milano, Italy). Purified saponins were examined as solutions in CD_3_OD (5 mg/mL), while condensed tannins as solution in CD_3_OD (10 mg/mL) in 5 mm tubes at 25 °C. ^1^H and ^13^C chemical shifts were expressed in ppm relative to TMS as standard.

## 4. Conclusions

Several species of *Hedysarum* have been investigated for their content of valuable phytochemicals [3,16], and the main attention has been given to the characterization of phenolics (flavonoids and tannins) and higher terpenes (saponins). In a previous study [21] on *H. coronarium*, only the content of condensed tannins and flavonoids in leaves was presented. To the best of our knowledge, this is the first chemical investigation of the flowers, in addition to the leaves, of this species. Moreover, in our study we have also characterized the saponin constituents of the two plant organs.

Similarly to previous data, our study confirmed that quercetin, kaempferol, and isorhamnetin (3-methoxyquercetin) are the dominant aglycones forming the different flavonoids; moreover, our study also confirmed that formononetin and afrormosin are the main aglycones present in the isoflavones derivatives. Compared to the work of Tibe et al. [21], we have detected in our material some more glycosyl derivatives of all the above aglycones. In particular, it is worth mentioning the identification of tricin-7-*O*-galactoside in the leaf extract, although in low amounts. Tricin is an *O*-methoxyflavone occurring in several plant species, especially cereal crop plants, as free or glycosylated aglycone, with a pest protective function [47].

Saponins are natural compounds widely distributed in legume plants [48,49] that differ quantitatively and qualitatively, both among and within species [48,50]. In agreement with data from a few other species of *Hedysarum* [3], soyasaponin glycosides have been detected as the only type of saponins in *H. coronarium* and may be considered a chemical marker for this genus.

Condensed tannins or proanthocyanidins are considered polyphenols with high molecular weights. The presence of multiple functional groups in their chemical structure, such as hydroxyls, provides them with the ability to create bonds to reach a stable cross-linked association within different molecules, such as proteins or carbohydrates. In particular, in forage legumes, prodelphinidins having more phenolic groups than procyanidins can better bind dietary proteins in animal rumen and improve protein utilization in ruminants. Furthermore, this type of condensed tannin has been reported to have higher biological activity than procyanidins [15]. Sulla tannins are prodelphinidin type tannins and can be considered valuable metabolites for different applications.

What is quite interesting is that our study indicated that flowers from sulla contain more than double the amount of flavonoids and saponins together with a higher amount of condensed tannins, supporting their edible use as an important source of healthy molecules [51]. All these groups of phytochemicals are well known for their biological and pharmacological effects [52,53,54,55], and their presence in *H. coronarium* suggests that a deeper investigation of their potential in human and animal health is required.

## Figures and Tables

**Figure 1 molecules-26-04606-f001:**
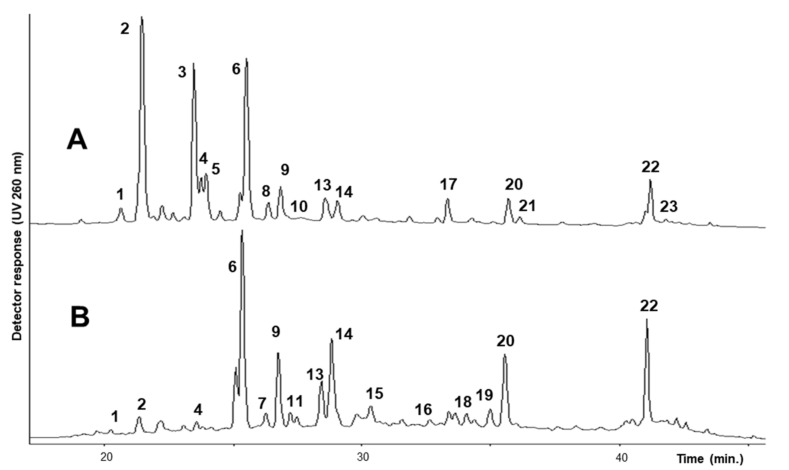
HPLC/DAD (UV 260 nm) chromatogram of flavonoids from (**A**) flowers and (**B**) leaves of *H. coronarium* L. For compounds numbering and identification, see Table 1.

**Figure 2 molecules-26-04606-f002:**
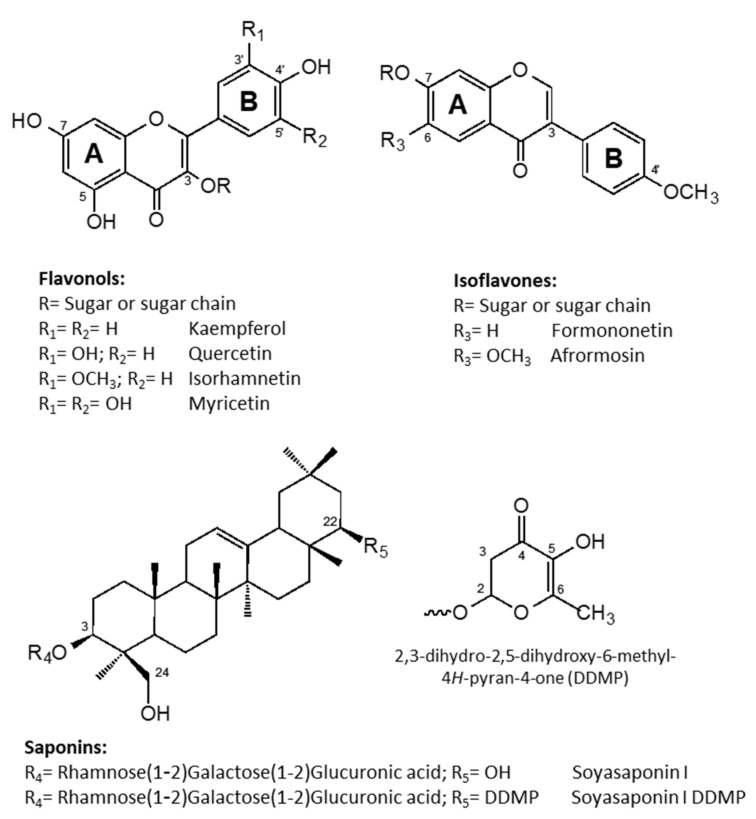
Chemical structures of the most abundant specialized metabolites detected in the flowers and leaves of *H. coronarium* L.

**Figure 3 molecules-26-04606-f003:**
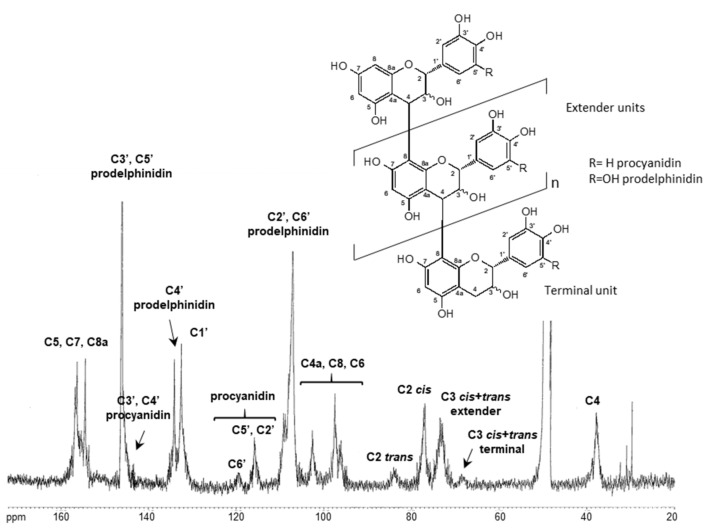
^13^C-NMR spectrum (CD_3_OD) of condensed tannins from *H. coronarium* L.

**Table 1 molecules-26-04606-t001:** Main identified flavonoids and their content (mg/g DM) in flowers and leaves of *H. coronarium* L.

#	UV l_max_(nm)	[M + H]^+^ (*m*/*z*)	[M − H]^−^ (*m*/*z*)	MW	Compound	Flowers	Leaves
**1**	255, 270*sh*, 358	627, 481, 319	625, 607, 317	626	Myricetin-3-*O*-rutinoside	0.11 ± 0.04	0.02 ± 0.01
**2**	254, 266*sh*, 300*sh*, 354	757, 611, 465, 449, 303	755, 737, 609, 591, 489, 301	756	Quercetin-3-*O*-(2″,6″-di-*O*-rhamnosyl)-glucoside	1.60 ± 0.38	0.13 ± 0.03
**3**	265, 299*sh*, 325*sh*, 354	741, 595, 449, 433, 287	739, 721, 593, 575, 393, 285	740	Kaempferol-3-*O*-(2″,6″-di-*O*-rhamnosyl)-glucoside	1.15 ± 0.12	-
**4**	256, 267*sh*, 354	611, 465, 303	609, 591, 301	610	Quercetin-3-*O*-rutinoside isomer	0.27 ± 0.05	0.03 ± 0.02
**5**	265, 298*sh*, 353	741, 595, 449, 433, 287	739, 593, 575, 285	740	Kaempferol-3-*O*-rutinoside-7*-O*-rhamnoside	0.33 ± 0.03	-
**6**	256, 266*sh*, 299*sh*, 356	611, 465, 303	609, 591, 301	610	Quercetin-3-*O*-rutinoside (rutin)	1.15 ± 0.23	0.71 ± 0.07
**7**	255, 260*sh*, 355	-	491, 329	492	Tricin-7-*O*-galactoside	-	0.05 ± 0.02
**8**	255, 266*sh*, 354	771, 625, 479, 463, 317	769, 623, 605, 315	770	Isorhamnetin-3-*O*-(2″,6″-di-O-rhamnosyl)-glucoside	0.12 ± 0.02	-
**9**	263, 299*sh*, 354	595, 449, 287	593, 285	594	Kaempferol-3-*O*-rutinoside (nicotiflorin)	0.26 ± 0.02	0.25 ± 0.04
**10**	265, 299*sh*, 353	-	447, 285	448	Kaempferol-3-*O*-galactoside	0.02 ± 0.01	-
**11**	256, 267*sh*, 354	551, 533, 465, 287	-	550	Quercetin-3-*O*-malonyl glucoside	-	0.06 ± 0.02
**12**	256, 267*sh*, 355	-	505, 463, 301	506	Quercetin-3-*O*-acetyl glucoside	-	0.04 ± 0.01
**13**	264, 300*sh*, 355	595, 449, 287	593, 285^-^	594	Kaempferol-3-*O*-rutinoside isomer	0.24 ± 0.04	0.14 ± 0.04
**14**	255, 266*sh*, 355	625, 479, 317	623, 315	624	Isorhamnetin-3-*O*-rutinoside	0.18 ± 0.01	0.38 ± 0.06
**15**	255, 266*sh*, 356	625, 479, 317	623, 315	624	Isorhamnetin-3-*O*-rutinoside isomer	0.05 ± 0.01	0.12 ± 0.06
**16**	254, 298*sh*, 357	-	655, 347	654	5,7-dihydroxy-3′,4′,5′-trimethoxyflavone-7-*O*-rutinoside	-	0.08 ± 0.03
**17**	243, 287, 318	-	517, 337, 179	518	Unidentified	0.21 ± 0.03	-
**18**	253, 355	-	507, 417, 387, 345	508	Unidentified	-	0.06 ± 0.03
**19**	258, 329	-	475, 267	476	Unidentified	-	0.06 ± 0.02
**20**	259, 355*sh*	431, 269	-	430	Formononetin-7-*O*-glucoside (ononin)	0.15 ± 0.06	0.22 ± 0.08
**21**	260, 320*sh*	461, 299	-	460	Afrormosin-7-*O*-glucoside	0.04 ± 0.02	-
**22**	260, 355*sh*	517, 431, 269	-	516	Formononetin-7-*O*-glucoside-6″-*O*-malonate	0.38 ± 0.06	0.37 ± 0.02
**23**	260, 320*sh*	547, 461, 299	-	546	Afrormosin-7-*O*-glucoside-6″-*O*-malonate	0.02 ± 0.00	-
					**Total**	**6.26 ± 0.75**	**2.72 ± 0.38**

**Table 2 molecules-26-04606-t002:** Saponin content (mg/g DM) in flowers and leaves of *H. coronarium*.

#		MW	Compound	Flowers	Leaves
**24**	C_48_H_78_O_18_	942	RhaGalGluA-SoyaB	3.79 ± 0.25	1.63 ± 0.17
**25**	C_42_H_68_O_14_	796	GalGluA-SoyaB	0.16 ± 0.02	0.02 ± 0.01
**26**	C_48_H_76_O_18_	940	RhaGalGluA-SoyaE	0.10 ± 0.02	0.14 ± 0.02
**27**	C_42_H_66_O_14_	794	GalGluA-SoyaE	0.41 ± 0.05	0.06 ± 0.01
**28**	C_54_H_84_O_21_	1068	RhaGalGluA-SoyaB-DDMP	1.74 ± 0.30	1.02 ± 0.06
**29**	C_48_H_74_O_17_	922	GalGluA-SoyaB-DDMP	0.09 ± 0.06	0.02 ± 0.01
			**Total**	**6.28 ± 0.30**	**2.88 ± 0.23**

**Table 3 molecules-26-04606-t003:** Thiolysis reaction products of total tannins and tannin fractions from *H. coronarium* as % contributors of terminal and extender flavan-3-ol units.

		% Terminal Units				% Extender Units				
	mDP	C	EC	GC	EGC	C	EC	GC	EGC	PD:PC
Total	21	57	7	30	5	5	7	32	56	88:12
Fraction 1	15	76	3	20	1	7	8	38	47	85:15
Fraction 2	25	51	19	19	11	4	7	33	57	90:10

## Data Availability

Data are contained within the manuscript.

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
