# Peer review of "Chemical Identification of Specialized Metabolites from Sulla (Hedysarum coronarium L.) Collected in Southern Italy"

_molecules, 2021, doi:10.3390/molecules26154606_

Round 1

Reviewer 1 Report

In this manuscript results from a detailed chemical analysis of the extracts from leaves and flowers of H. coronarium grown wild in southern Italy are reported. Identification of the main specialized metabolites within the chemical classes of flavonoids, proanthocyanidins and saponins is described including considerations on their content in the two plant organs. Information acquired from this study expands the knowledge on H. coronarium as a source of valuable phytochemicals for different applications in human and animal health and nutrition.

Overall, the study is well-designed and well-written, the methods are suitable and the obtained results are clearly presented and discussed.  Moreover, the conclusions have been appropriately pointed out.
The content of this manuscript matches well with the journal’s purpose, but some clarifications are needed:
line 148-15: "-O-" should be italic
line 266: copmared with? could you explained?
line 361: m/z should be italic

Author Response

We thank Reviewer 1 for helpful and valuable comments. All the corrections are inserted in the text.

Reviewer 2 Report

The subject of the manuscript is very interesting and with possible practical implementation. Information acquired from this study expands the knowledge on H. coronarium as a source  of valuable phytochemicals for different applications in human and animal health and nutrition. I would recommend the publication of this work after listed issues are addressed.

General Comment:

This work is missing information how all methods were tested to ensure proper results are obtained. At least basic method transfer is needed to check specificity, range, repeatability etc. I believe such tests were performed but not described. This is especially important for HPLC methods. A summary of these activities should be included.

Specific Comments:

  1. 379-384 I recommend that you describe the calibration curves. In accordance with the General Note, the reliability of the method of analysis should be confirmed.
  2. 387 I recommend describing the method.
  3. 428 In accordance with the General Note, the reliability of the method of analysis should be confirmed.

Author Response

We thank Reviewer 2 for helpful and valuable comments. All the requested information are now inserted in the text.

Reviewer 3 Report

This manuscript is well written. English errors are minor. Results are presented in a clear manner. The only drawback is the lack of important references that the authors did not discuss. I suggest acceptance of this paper after its major revision, i.e. authors should discuss and cite the following references:

Burlando, B., Pastorino, G., Salis, A., Damonte, G., Clericuzio, M., & Cornara, L. (2017). The bioactivity of Hedysarum coronarium extracts on skin enzymes and cells correlates with phenolic content. Pharmaceutical Biology, 55(1), 1984-1991. doi:10.1080/13880209.2017.1346691

Jerkovic, I., Tuberoso, C. I. G., Gugic, M., & Bubalo, D. (2013). Composition of Sulla (Hedysarum coronarium L.) Honey Solvent Extractives Determined by GC/MS: Norisoprenoids and Other Volatile Organic Compounds (vol 15, pg 6375, 2010). Molecules, 18(11), 13434-13434. doi:10.3390/molecules181113434

Sulas, L., Campesi, G., Piluzza, G., Re, G. A., Deligios, P. A., Ledda, L., & Canu, S. (2019). Inoculation and N Fertilization Affect the Dry Matter, N Fixation, and Bioactive Compounds in Sulla Leaves. Agronomy-Basel, 9(6). doi:10.3390/agronomy9060289

Author Response

We thank Reviewer 3 for helpful and valuable comments. We have added two of the suggested references by Referee 3. As regards the paper of Jerkovic et al., we believe that its subject "extractives of sulla honey" is not relevant in the context of our data. If the Editor agrees, we would like not to cite it.

Round 2

Reviewer 3 Report

I suggest acceptance of this manuscript.